# DGRPool, a web tool leveraging harmonized *Drosophila* Genetic Reference Panel phenotyping data for the study of complex traits

Vincent Gardeux[1,2], Roel PJ Bevers[1†], Fabrice PA David[2,3], Emily Rosschaert[1,4], Romain Rochepeau[1], Bart Deplancke[1,2]*

[1]Laboratory of Systems Biology and Genetics, Institute of Bioengineering, School of Life Sciences, Ecole Polytechnique Fédérale de Lausanne (EPFL), Lausanne, Switzerland; [2]Swiss Institute of Bioinformatics, Lausanne, Switzerland; [3]Bioinformatics Competence Center, EPFL, Lausanne, Switzerland; [4]Laboratory of Behavioral and Developmental Genetics, Center for Human Genetics, KU Leuven, Leuven, Belgium

*For correspondence: bart.deplancke@epfl.ch

Present address: †Genomics England Ltd., London, United Kingdom

Competing interest: The authors declare that no competing interests exist.

**Abstract** Genome-wide association studies have advanced our understanding of complex traits, but studying how a GWAS variant can affect a specific trait in the human population remains challenging due to environmental variability. *Drosophila melanogaster* is in this regard an excellent model organism for studying the relationship between genetic and phenotypic variation due to its simple handling, standardized growth conditions, low cost, and short lifespan. The *Drosophila* Genetic Reference Panel (DGRP) in particular has been a valuable tool for studying complex traits, but proper harmonization and indexing of DGRP phenotyping data is necessary to fully capitalize on this resource. To address this, we created a web tool called *DGRPool* (dgrpool.epfl.ch), which aggregates phenotyping data of 1034 phenotypes across 135 DGRP studies in a common environment. DGRPool enables users to download data and run various tools such as genome-wide (GWAS) and phenome-wide (PheWAS) association studies. As a proof-of-concept, DGRPool was used to study the longevity phenotype and uncovered both established and unexpected correlations with other phenotypes such as locomotor activity, starvation resistance, desiccation survival, and oxidative stress resistance. DGRPool has the potential to facilitate new genetic and molecular insights of complex traits in *Drosophila* and serve as a valuable, interactive tool for the scientific community.

## eLife Assessment

Genetic analysis of complex traits in *Drosophila* provides a resource for exploring the relationship between genetic and phenotypic variation. The web tool DGRPool presented in this paper makes data and results from the *Drosophila* Genetic Reference Panel accessible that will enable downstream analyses of genetic association. The findings of this paper are considered to be **important**, with practical implications beyond a single subfield, supported by **convincing** evidence using appropriate and validated methodology in line with current state of the art.

## Introduction

*Drosophila melanogaster* is an excellent model organism for studying genotype-to-phenotype relationships. It is a short-living species and is very easy to maintain in similar laboratory conditions, which limits confounding factors such as the environment. The *Drosophila* Genetic Reference Panel

(DGRP) was created in the early 2010s and now consists of 205 inbred lines that are fully sequenced, of which 192 are still available in the Bloomington *Drosophila* Stock Center (https://bdsc.indiana.edu/; *Mackay, 2012*; *Huang et al., 2014*). The DGRP has proven highly valuable to study the genetic basis of complex traits, as illustrated by the many studies that have used GWAS principles to identify variants that contribute to traits related to morphology, metabolism, behavior, aging, disease susceptibility etc. (*Figure 1A*). Furthermore, since the DGRP lines were inbred for many generations, they are almost fully homozygous, which simplifies the identification of putatively causal alleles and elucidation of implicated molecular mechanisms (*Bou Sleiman et al., 2015*). Moreover, the fact that the same lines can be studied by various researchers for diverse traits should leverage these data generation efforts to uncover unexpected correlations between phenotypes or relationships between genetic variants and a wide range of traits.

However, there is currently only one major data resource that aims to compile DGRP information, the DGRP2 website (http://dgrp2.gnets.ncsu.edu/; *Mackay, 2012*; *Huang et al., 2014*). This website hosts the genotyping data, its annotation, and potential known covariates (well-established inversion genotypes and symbiont levels), as well as 31 phenotypes from 12 studies (*Table 1*). The data is primarily hosted as static files, downloadable from the website, along with limited RNA expression data. In addition, a very important tool, used by the DGRP community, is the possibility for any user to submit their own phenotype files for running a GWAS analysis (corrected for the so-called 'known covariates', i.e. established inversions and *Wolbachia* infection status). This is particularly useful, especially for researchers that do not have the bioinformatics knowledge or capacity to perform these tasks internally. However, the DGRP2 website has not been updated for an extended period as the last referenced paper dates back to 2015, and, except for the GWAS computation, has remained static. This means that any meta-study, which would aim to aggregate datasets across available phenotypes, would require hours (if not days) of work to transform the data into an appropriate and common format. Moreover, the result of such effort would unlikely become available to the rest of the community, and thus any other group would need to redo this work in order to gather similar information, while the data of other phenotyping studies beyond the 12 available would not be easily accessible.

For all these reasons, we decided to create a web application that would both act as a repository of DGRP phenotyping datasets and also as an online tool for assisting researchers with some basic systems genetics-inspired analyses. Our goal was to index all existing literature about DGRP phenotyping data —where possible— in order for users to quickly search through the website using simple keywords. We manually associated each study with broad and tailored categories such as 'ageing', 'metabolism', or 'olfactory'. We specifically spent important time curating the datasets to avoid any errors or misrepresentations of datasets. To avoid the 'maintenance issue' that is common to online tools, and keep the data up to date, we implemented specific curators' tools, to help maintain the web application in the future. These tools allow any user to submit a novel dataset, which is then attributed to a curator, in order to manually format and validate all phenotyping data and metadata associated with the study. Importantly, any user can become a curator, as advertised on the main page of the resource, since we strongly believe that such a community-run resource architecture is most optimal to keep a web tool state-of-the-art and allow crowd-based curation work (*Gramates et al., 2022*).

In addition, we set out to build important tools for the DGRP community such that DGRPool would not only be a static repository for downloading phenotyping data but could also be used as an interactive data analysis tool. For example, users can correlate phenotypes together, from the same study or across studies. We also implemented an automated GWAS analysis (using PLINK2, and known covariates) which we pre-calculated on all the phenotyping data that are currently available. Using this data, users can simply browse through their genes or variants of interest and directly find related phenotypes. A PheWAS page also allows exploration of each variant's impact across multiple phenotypes. Moreover, these tools are applicable to user-submitted phenotypes, so that anyone can upload their own phenotypes to search the DGRPool database for correlated phenotypes or to run GWAS analyses.

Our goal is to ensure that DGRP phenotyping data is findable, accessible, interoperable, and reusable (FAIR) (*Wilkinson et al., 2016*) to fully leverage the opportunities that stem from this unique genotyping-phenotyping resource. To this end, we made user access our priority, both for removing the bottleneck of data harmonization, and also to allow for better, more reproducible research. This

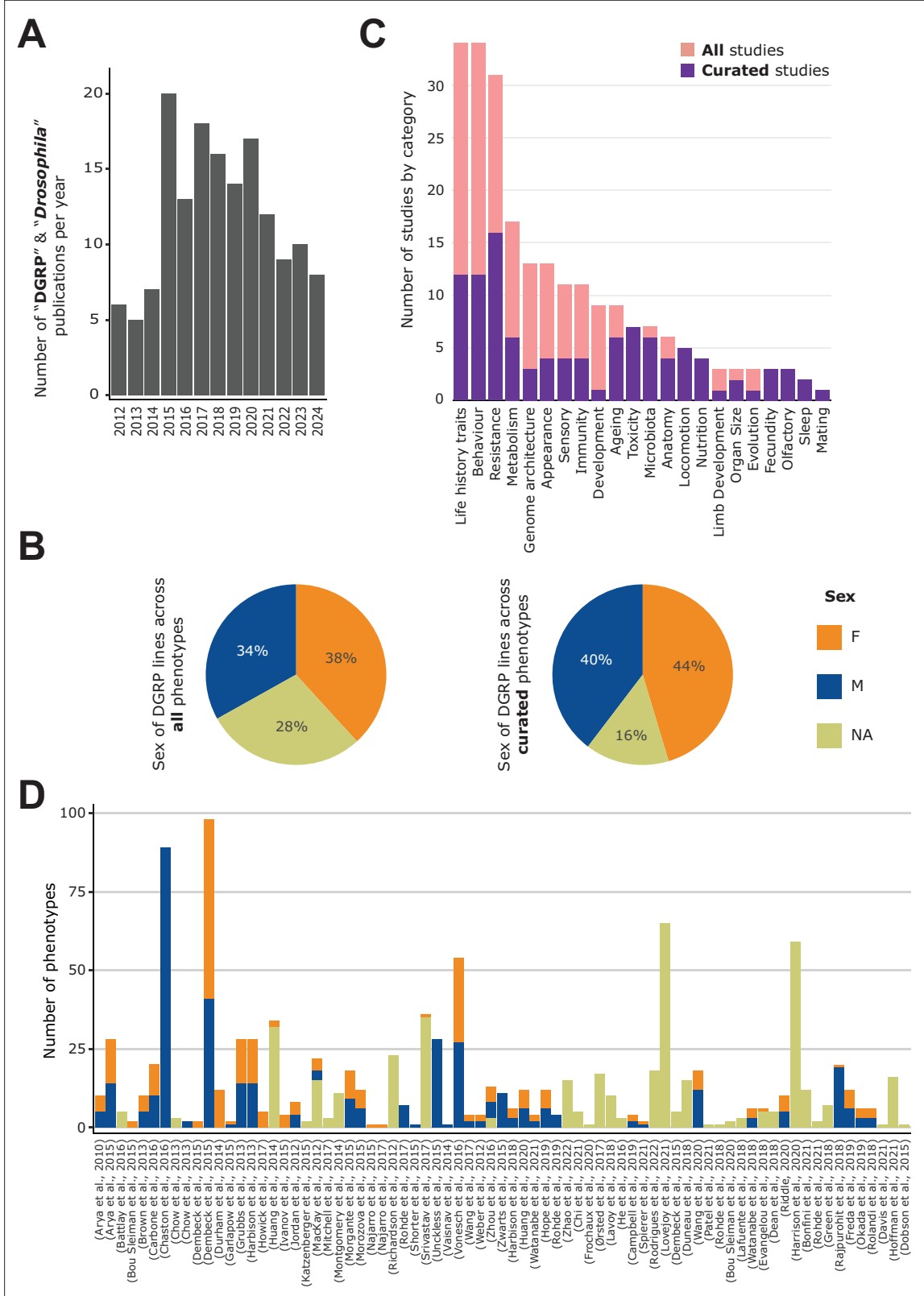

**Figure 1.** General content of the DGRPool web tool. (**A**) Pubmed search on 'Drosophila DGRP' terms unveiled 155 results from 2012–2024 (search made in July 2024). (**B**) Sex of the DGRP lines used across all 135 studies (left) and 43 curated studies (right), for each phenotype. (**C**) Number of studies per phenotype category. Studies can be assigned to multiple categories. (**D**) Number of phenotypes per study and per sex. Studies without attached phenotypes were not plotted. Of note, a given phenotype can be measured for different sexes and thus counted multiple times.

*Figure 1 continued on next page*

*Figure 1 continued*

The online version of this article includes the following figure supplement(s) for figure 1:

**Figure supplement 1.** Screenshot from the curator's view for a given study - Metadata section.

**Figure supplement 2.** Number of phenotypes per study.

**Figure supplement 3.** Screenshot from the curator's view for a given study - Phenotype section.

aligns with the harmonization effort undertaken by other human GWAS/PheWAS resources, such as the GWAS Catalog (*Sollis et al., 2023*), Open Targets Genetics (*Ghoussaini et al., 2021*), MR-base (*Hemani et al., 2018*), and the FinnGen portal (*Kurki et al., 2023*), which provide extensive examples of effective data use and accessibility. Although the structure of DGRPool differs from these human

**Table 1.** Comparison of the two currently available web portals organizing DGRP phenotyping data. This table compares different features available in DGRPool, with DGRP2 being the main current resource for DGRP data. DGRPool separates the features into (1) *Data*, which summarizes the available phenotyping data, (2) *Tools*, which lists the available tools and options, mainly GWAS, PheWAS and phenotype correlation, (3) *Web*, which describes the website itself, and (4) *Additional features* that are available in DGRPool, such as the curation system, the possibility to publish new studies and the interactive plots. Of note, the 838 phenotypes are counted regardless of the linked sex (M, F and/or NA), while the 'sex-specific' value is calculated by counting the same phenotype separately for each available sex.

| | | | DGRPool | DGRP2 |
|---|---|---|---|---|
| REFERENCE | | | This study | *Mackay, 2012*; *Huang et al., 2014* |
| DATA | | *DGRP lines* | 342 | 205 |
| | | *DGRP studies* | 135 (43 fully curated) | 12 |
| | | *Phenotypes* | 1034 (840 unique) | 31 |
| | | *Gene Expression data* | External links | ✓ |
| TOOLS | GWAS | *Calculated on all phenotypes* | ✓ | |
| | | *User upload* | ✓ | ✓ |
| | | *Method* | Plink2 | FastLMM |
| | | *Covariates* | Wolbachia + 5 Insertions | Wolbachia + 5 Insertions |
| | | *Boxplot of REF vs ALT* | ✓ | |
| | | *PheWAS of top variants* | ✓ | |
| | *Phenotype correlation* | *Calculated on all phenotypes* | ✓ | |
| | | *User upload* | ✓ | |
| WEB | | *URL* | https://dgrpool.epfl.ch/ | http://dgrp2.gnets.ncsu.edu/ |
| | | *Backend* | Ruby-on-rails+PostgreSQL | NA |
| | | *Frontend* | Javascript, Plotly | NA |
| FEAT. | | Curation system & tools | ✓ | |
| | | Publish new studies | ✓ | |
| | | Interactive plots | ✓ | |

databases, we acknowledge the importance of similar meta-data harmonization guidelines. Inspired by the GWAS Catalog's summary statistics submission guidelines, we propose submission guidelines for DGRP phenotyping data in this paper.

To showcase the potential of our tool in facilitating new biological discoveries, we conducted a proof-of-concept study focusing on the longevity phenotype, a well-studied trait in *Drosophila* research with clear relevance to human longevity (*Piper and Partridge, 2018*). By leveraging the data harmonization and curation efforts in DGRPool, we identified multiple phenotypes that are significantly associated with longevity across 18 different studies, such as oxidative stress resistance (*Finkel and Holbrook, 2000*), sleep duration (*Bushey et al., 2010*; *Thompson et al., 2020*), desiccation survival (*Rion and Kawecki, 2007*; *Hoffmann and Harshman, 1999*), and starvation resistance (*Rion and Kawecki, 2007*; *Chippindale et al., 1996*; *Jang and Lee, 2015*). Interestingly, we also observed phenotype-phenotype correlations between 'shorter lifespan' and certain phenotypes, such as locomotor activity (*Magwere et al., 2006*) and food intake (*Lee et al., 2008*; *Piper and Partridge, 2007*). These results validate prior knowledge and illustrate how our tool can provide novel biological insights with just a few clicks. Therefore, we firmly believe that tools such as DGRPool —which ultimately could become entirely community-driven— are essential not only for catalyzing novel research, but also for leveraging the diversity and richness of existing datasets.

## Results

### A thousand phenotypes across 135 studies

To start our data collection, we searched for DGRP studies that reference any phenotyping data and, in parallel, implemented diverse tools to automatically aggregate these data and their associated metadata from the journals hosting the datasets. However, we quickly realized that it was difficult to automate the entire process. Specifically, the import of phenotyping data proved challenging since (i) datasets tended to be hosted in very different formats such as Excel files or PDF, (ii) data was stored within the journal's supplementary section, or in external repositories such as Figshare; and (iii) the format of the phenotyping data differed from one publication to another. Because of these challenges, we implemented a curation page to manually review, edit, and correct datasets that were automatically aggregated, aiming to prevent errors in the imported datasets. In addition, this allows the curator to add relevant remarks or comments on the study under review, thus providing enhanced context for future analyses of these datasets.

In line with the community-resourcing philosophy of DGRPool, we created a specific 'curator' role that any logged-in user can claim, again with the underlying rationale of assuring long-term sustainability of our web application. With this role, the user has access to additional functionalities on the DGRPool website, including the modification of any metadata attached to a study (title, authors, description, categories), and the submission or modification of attached phenotypes (*Figure 1—figure supplement 1*). Although this may entail a considerable amount of time, we assert that this approach is the most effective means of furnishing high-quality data. Consistent with this philosophy, we have incorporated a functionality on the homepage which empowers any user to submit a DOI as a recommendation for a study that could be absent from the DGRPool repository. If the DOI is not in the database, it triggers the same automated scripts that were originally used to incorporate the 135 studies. The corresponding study is then created on DGRPool, and its metadata (authors, links, …) are automatically imported. Once a study has been created, one of three possible labels can be assigned to describe the state of curation of a study: (1) *Submitted* (default), when no curator is yet assigned to the study, (2) *Under curation*, when a curator is assigned, and (3) *Curated* when all phenotyping data and metadata have been curated, and the study received final approval by the curator. At this time, DGRPool hosts 135 studies, including 43 that have already been fully curated, 80 still under curation, and 12 under a submitted status (i.e. the DOI was submitted by users as a relevant DGRP study not yet present in the database). In total, 75 studies have attached phenotyping data; 100% of the curated ones, and only 40% of the non-curated ones. Altogether, the total number of studies in DGRPool is currently 135, and we expect that this number will continue to grow upon its public release, along with the number of curated studies.

Since the curation process is still ongoing, we will be referring to two different datasets in the manuscript: (1) The *full dataset*, comprising *135* studies (independent of 'curation' status), and (2) the

*curated dataset*, comprising *43* studies that have already underwent thorough curation and contributed about 505 phenotypes (see below). Of note, for all tools available on the website, it is possible to run these on either all studies or (as is currently the default), only on the curated studies.

For all of the curated studies, we carefully separated the data by sex when information on sex-specific phenotypes was available, or we assigned it as *NA* when flies were sex-mixed, when there was no information on sex, or when the phenotype is inherent to a population (e.g. in the case of non-sexual chromosomal traits, like inversions). We also extracted this information from the phenotyping data itself for the non-curated studies, when available, but when not findable, it was set to *NA*, waiting for a more in-depth curation and careful reading of the paper's method section. Therefore, across all 135 studies, this led to an overall equilibrium between all represented sexes, with slightly more data for females and slightly less unannotated data (*Figure 1B*). However, when focusing only on the 43 curated datasets, the proportion of phenotypes without assigned sex (*NA*) dropped drastically to ~15%. This effect highlights the importance of rigorous curation, which typically requires the curator to read through the entire manuscript to understand the utilized experimental protocols to select the appropriate sex, even if this information is not explicitly indicated in the phenotyping data itself.

Upon data curation, the assigned curator(s) has to specify a few phenotypic categories for each study, for example, 'Metabolism', 'Nutrition', or 'Ageing' (*Figure 1C*). Since these categories are browsable, it facilitates searching for a set of specific studies or linking the studies together. Interestingly, the top annotated categories are either 'Behaviour', 'Life History Traits', or 'Resistance', which is consistent with historical behavioral and immune studies conducted for *Drosophila* as a model organism (*Arch et al., 2022*; *Dissel, 2020*; *Flatt, 2020*; *Harnish et al., 2021*; *O'Kane, 2011*). The number of phenotypes per study ranges from 1 to 89 (*Figure 1D*, *Figure 1—figure supplement 2*), with a median of 5, and a mean of 11, revealing that, while a low number of phenotypes (usually less than 10) tends to be the norm, some studies aggregate lots of (often similar) phenotypes. An example of the latter is *Chaston et al., 2016* which investigated the impact of microbiota on nutritional traits. The authors studied 76 different microbial taxa, whose effect was quantified independently, generating a high number of phenotypes. Similarly, *Dembeck et al., 2015* studied cuticular hydrocarbon composition, considering 66 different cuticular components, while *Vonesch et al., 2016* studied organismal size traits, regrouping 28 morphological phenotypes such as wing length or intraocular distance. In total, the 43 curated studies aggregate 316 M+220 F+133 NA=669 sex-specific phenotypes, for a total of 505 unique phenotypes (~60%), while the remaining non-curated studies provide another 60 M+37 F+268 NA=365 sex-specific phenotypes, for a total of 333 unique phenotypes (~40%).

## Harmonization and formatting of phenotyping data

DGRP phenotyping data are often available as a supplemental data table, published along with the main paper on the journal's website. Such data can also be stored on external websites such as Figshare and, as already indicated, the corresponding file can be in varying formats (i.e. Excel, text, or PDF), so it is challenging to entirely automate extraction algorithms. Usually, the data are presented in the form of a matrix, with DGRP lines in rows and phenotypes in columns. But sometimes, they can be in a more 'exotic' format (*Hope et al., 2019*), requiring a hands-on approach to format it appropriately. Also, the provided phenotyping data are often not sufficiently self-informative and thus require in-depth reading of the original manuscript to grasp abbreviations or identify the correct measurement units. These are important, in particular, to assure reproducibility, but especially when aggregating multiple studies together such that the scale of the values is similar. In DGRPool, we therefore created a common matrix format to represent all studies, and we implemented a 'Unit' metadata for each phenotype. Then, for each study, we mapped all phenotypes to their appropriate format and units (*Figure 1—figure supplement 3*). This part is fully accessible to the curator, who can update or add any phenotype that would be missing, with their corresponding units and meta-data description.

Another issue that we faced is that phenotypes are often averaged across multiple individual flies and that the authors only provide these 'Summary datasets'. This can be problematic in terms of reproducibility, since some figures may show boxplots or distributions of values for each DGRP line, but these plots are not reproducible when only summary data is available (i.e. means or medians). Fortunately, some studies do provide 'raw datasets' which contain multiple phenotypic values per

DGRP line, often corresponding to replicate flies of the same genotype. These values tend to be of much greater interest since they enable statistical analyses and/or the computation of further summary statistics (not only mean or median, but also the standard error of means or other often non-provided summary values).

Finally, for some studies, phenotyping data were not or no longer available from the journal's website (*Battlay et al., 2016*; *Durham et al., 2014*; *Najarro et al., 2017*), which is often the journal's responsibility. However, in all cases, we were able to contact the authors directly to recover the missing datasets.

To avoid such issues in the future, we have formulated a couple of good practice guidelines for authors to facilitate and improve upon our and future datasets with the aim of enabling harmonized and reproducible research. These guidelines are detailed in the Discussion section of this manuscript. All curated datasets in DGRPool are formatted following these guidelines (where possible), and phenotypes can now be easily downloaded in a standard TSV format from a particular study, or from a phenotype page.

## How to leverage these datasets by correlating phenotypes

Our formatting and harmonizing of all datasets now enable interesting cross-phenotype analyses to generate new biological insights. One strategy to perform such analyses is to download a summary table that contains all the phenotypes in a common format and that is available from DGRPool's front page. However, we deemed this still insufficient as a catalyzing resource, which is why we implemented tools to correlate existing and user-submitted phenotypes with all the other phenotypes in DGRPool (*Figure 2—figure supplement 1*).

To better understand the structure of these phenotypes, and how they relate together, we also computed a global visualization of the phenotype correlations across all curated studies (*Figure 2A*, *Figure 2—figure supplement 2*). Of note, by 'phenotype correlations', we mean direct phenotype-phenotype correlations, that is, a straightforward Spearman's correlation of two phenotypes between common DRGP lines, and we repeated this process for each pair of phenotypes. This revealed a clear trend, with phenotypes belonging to the same study (within-study) correlating in general stronger than those from different studies (*Figure 2B*, *Figure 2—figure supplement 3*). This is expected since a given study will typically contain phenotypes that have been acquired for a given research topic, thus they will share similarities. Another potential factor that could explain this similarity is the well-known 'batch effect'. Indeed, phenotypes acquired in the same environment (same lab, technician, reagents etc.) may sometimes show greater similarity than those acquired across different labs and conditions (*Ackermann, 2001*). The longevity phenotype however, assessed in at least six of the studies in DGRPool (*Durham et al., 2014*; *Arya et al., 2010*; *Huang et al., 2020*; *Ivanov et al., 2015*; *Zhao et al., 2022*; *Hoffman et al., 2021*) across different laboratories, illustrates that phenotype and its measurements not only exhibits strong correlation across sexes (*Figure 2C*), but are also sufficiently robust between laboratories (*Figure 2D*). This example illustrates both the high robustness of results acquired in the context of DGRP studies (stable genotype, stable environment) and the robustness of the phenotype itself, which highlights its potential high heritability.

## Cross-study correlations highlight phenotype relationships

*Figure 2A* also highlights interesting cross-study correlations. For example, we can see a strong correlation between *Vonesch et al., 2016* and *Grubbs et al., 2013* which is perhaps expected since both studies examine fly morphology traits. The first one measures different organismal size traits such as eye interocular distance, or wing length, while the second studies leg and antenna development from imaginal discs, resulting in measuring phenotypes such as leg and bone length (*Figure 3A*). Similarly, three studies: *Mackay, 2012*, *Richardson et al., 2012*, and *Huang et al., 2014* are expectedly correlated since all three investigate the influence of the *Wolbachia* endosymbiont. Another interesting correlation is between *Chow et al., 2013* and *Durham et al., 2014* which both studied fecundity and yield a cross-study correlation between remating proportion *Chow et al., 2013* vs. mean fecundity *Durham et al., 2014*; *Figure 3B*. While potentially conceptually obvious, this correlation suggests that females that are more likely to mate with multiple males tend to also produce a greater number of eggs.

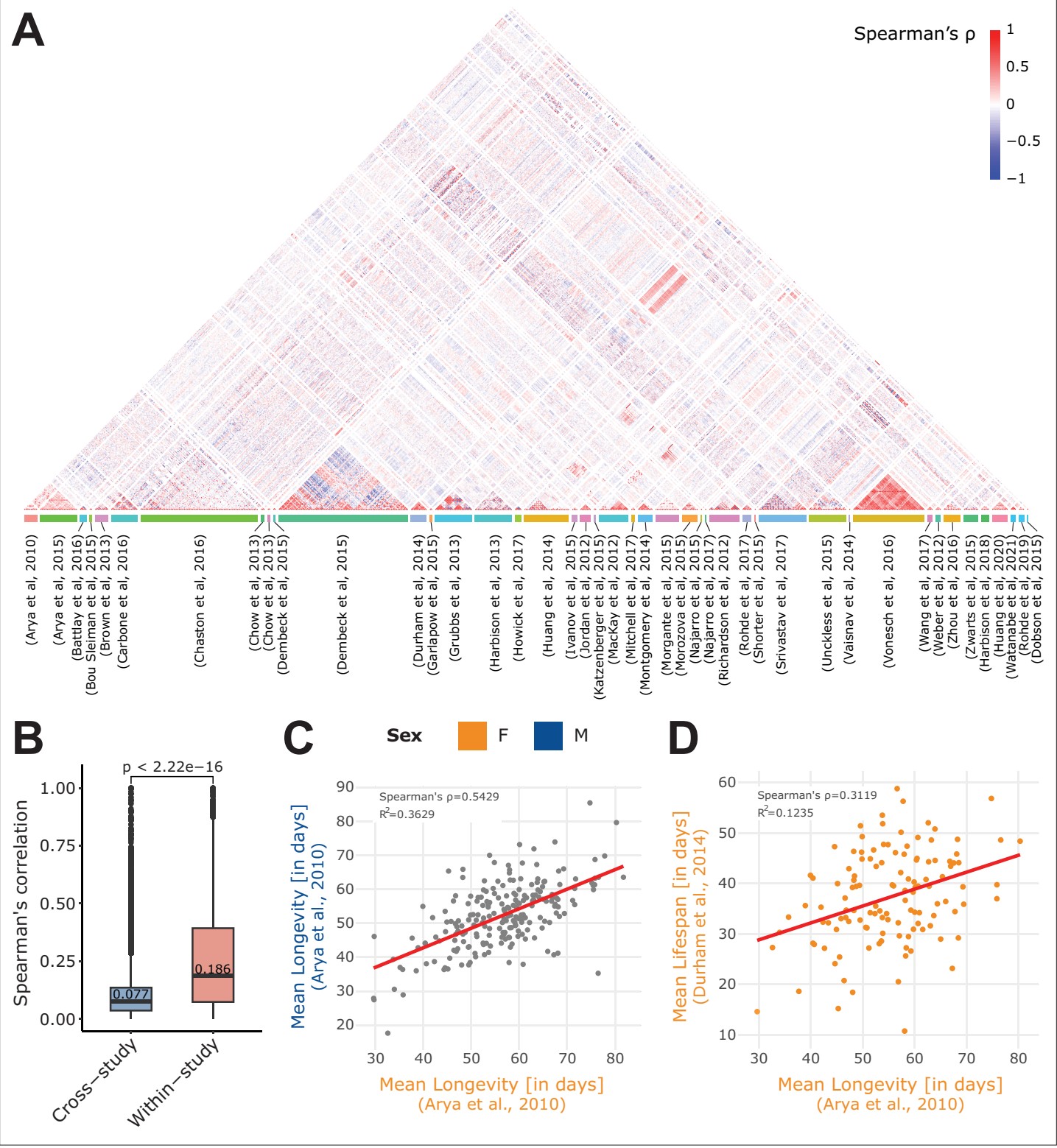

**Figure 2.** Within- and cross-study phenotype correlations. (**A**) Spearman's correlation of all phenotypes available in the 43 curated studies. Of note, we separately computed the phenotype correlations when data per sex were available (M, F, or NA), and we restricted the computation to quantitative (non-categorical) phenotypes. Phenotypes are grouped by study (colored box at the bottom of the plot). (**B**) Absolute value of the Spearman's correlation of pairs of phenotypes that originated from the same study (within-study) and those that originated from two different studies (cross-study). Of note, displayed values are median. Mean values are 0.099 for cross-study, and 0.259 for within-study. Again, we restricted the calculation to the 43

*Figure 2 continued on next page*

*Figure 2 continued*

curated studies. (**C**) Correlation of two longevity phenotypes from the same study (**Arya et al., 2010**), revealing a strong correlation between Female (**F**) and Male (**M**) longevity. (**D**) Correlation of two phenotypes from different studies: mean lifespan (**Durham et al., 2014**) and mean longevity (**Arya et al., 2010**). Of note, both the C and D plots were generated using the 'phenotype correlation' tool in DGRPool.

The online version of this article includes the following figure supplement(s) for figure 2:

**Figure supplement 1.** Screenshot from the phenotype correlation tool result page.

**Figure supplement 2.** Spearman's correlation of all phenotypes available in the 43 curated studies.

**Figure supplement 3.** Comparison of correlation within and cross-study.

These examples were all generated using the DGRPool phenotype correlation tool, supporting our notion that it can leverage cross-study comparisons of multiple phenotypes to unveil potentially new interesting phenotype interaction/associations. As a further proof of concept and given society's strong interest in defining 'healthy aging' determinants (**Friedman, 2020**), we continued investigating the 'mean longevity' phenotype from **Arya et al., 2010** and we studied the 33 phenotypes that were significantly correlated with it at 5% FDR threshold (**Figure 3C**). The hierarchical clustering clearly separated the phenotypes into three clusters: longevity-like phenotypes (strongly correlated together), other longevity-associated phenotypes (correlated with longevity), and phenotypes that seem antagonistic to longevity (anti-correlated phenotypes). Among the phenotypes that positively correlated with longevity, some may be expected such as starvation resistance (**Rion and Kawecki, 2007**; **Chippindale et al., 1996**; **Jang and Lee, 2015**) and oxidative stress resistance (**Finkel and Holbrook, 2000**) but some are less intuitive such as desiccation survival (**Rion and Kawecki, 2007**; **Hoffmann and Harshman, 1999**), and certain cuticular components of the epicuticle (**Wang et al., 2022**). We further investigated the results for an alternative Pearson's correlation test, at 25% FDR threshold (**Figure 3—figure supplement 1**), and recapitulated most of these findings, with the addition of sleep duration (**Bushey et al., 2010**; **Thompson et al., 2020**), whose relationship to longevity is complex and still not fully understood (**Watson et al., 2015**). Although we cannot exclude spurious correlations, some of these more surprising correlations appear biologically highly interesting, illustrating the capacity of DGRPool to unveil new research avenues that seem worth exploring in greater molecular detail. Also of interest is the group of often unexpected phenotypes that significantly anti-correlates with longevity. These include locomotor activity (**Magwere et al., 2006**), and some other cuticular components of the epicuticle (**Nghiem et al., 2000**), suggesting that higher locomotor activity is linked to reduced longevity. At 25% FDR (**Figure 3—figure supplement 1**), we can also see an anti-correlation with food intake (**Lee et al., 2008**; **Piper and Partridge, 2007**), which potentially recapitulates previous results stating that limiting food intake increases organismal lifespan (**McCracken et al., 2020**). Whether these are direct or indirect links remains unanswered, but appears worthy for a more in-depth scrutiny that is beyond the scope of this paper.

Inversely, our analyses also revealed that some expected phenotype correlations could not be detected. For example, in the context of metabolic energy expenditure (**Chatterjee and Perrimon, 2021**), it might seem intuitive that higher activity (**Harbison et al., 2013**) would lead to greater food intake (**Garlapow et al., 2015**). However, we did not observe such a correlation. Similarly, higher activity levels may reflect increased mating behaviour (**Chow et al., 2013**), but this was also not observed. These are just a few examples of several cases where expected correlations did not materialize, collectively signifying that the genetic architecture underlying such traits appears inherently complex.

These proof-of-concept examples demonstrate in our opinion the utility of the DGRP lines and by extension DGRPool to serve as powerful tools that will facilitate the identification of non-intuitive phenotype correlations and their underlying molecular basis as well as the discovery of putative genotype to phenotype relationships, as detailed below.

## From phenotypes to associated genotypes

The goal of most DGRP phenotyping studies is to eventually be able to link the phenotypes to potentially causal variants or sets of variants (**Mackay and Huang, 2018**). In response, tools like DGRP2 GWAS http://dgrp2.gnets.ncsu.edu/; **Mackay, 2012**; **Huang et al., 2014** have been put in place to accommodate geno-phenotype relationship analyses.

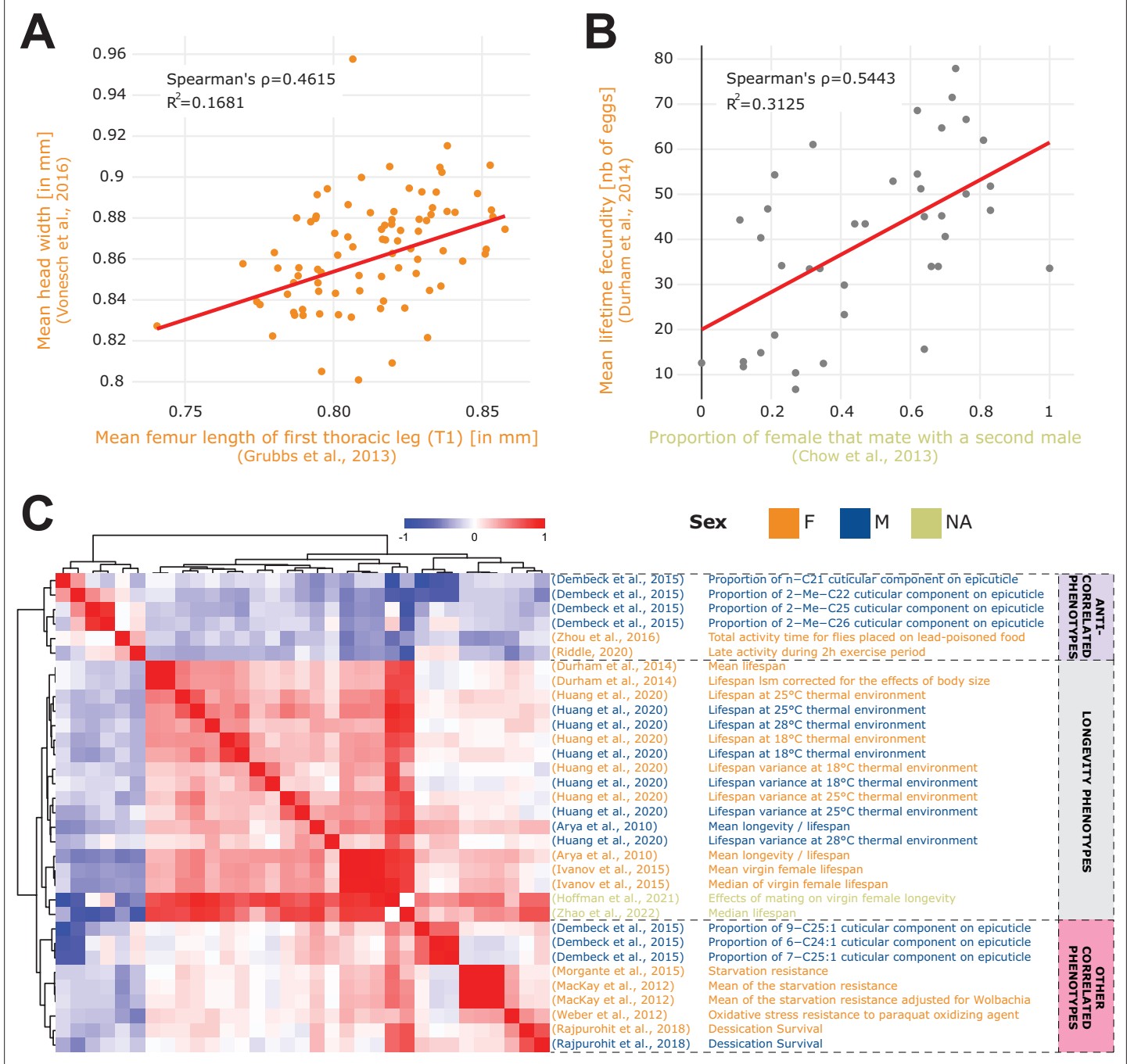

**Figure 3.** Phenotype correlations contribute new biological insights. (**A**) Correlation of mean femur length *Grubbs et al., 2013* vs. mean head width *Vonesch et al., 2016* showing the significant cross-study association of organismal size traits. (**B**) Correlation of remating proportion *Chow et al., 2013* vs. mean fecundity *Durham et al., 2014*. (**C**) 33 phenotypes correlated with longevity *Arya et al., 2010* at a 5% FDR threshold (Spearman's correlation), revealing three main groups of phenotypes: lifespan phenotypes (middle rows), other correlated phenotypes (bottom rows) and anti-correlated phenotypes (top rows). Of note, both the A and B plots were generated using the 'phenotype correlation' tool in DGRPool.

The online version of this article includes the following figure supplement(s) for figure 3:

**Figure supplement 1.** Phenotype correlations contribute new biological insights.

With the goal of providing an integrative analytical environment, we therefore also implemented GWAS tools within DGRPool (https://dgrpool.epfl.ch/check_pheno), aiming to assist researchers with performing GWAS analyses and interpreting the respective output. Specifically, we precalculated GWAS analyzes using PLINK2 on every existing phenotype in DGRPool (see Methods), thereby considering all ~4 M available DGRP variants while correcting for six known covariates (*Wolbachia* status, and five major insertions; *Huang et al., 2014*). Consequently, users can browse the GWAS results from any phenotype page on DGRPool (*Figure 4—figure supplement 1*). These comprise a QQplot, for assessing the validity of the results, or potentially over-estimated p-values, and a Manhattan plot, for visualizing the significant loci across the *D. melanogaster* genome. It also displays a table with the top 1000 associated variants and allows the user to download the table of all significant hits, at a p-value <0.001 threshold. The tool further runs a gene set enrichment analysis of the results filtered at p<0.001 to enrich the associated genes to gene ontology terms, and Flybase phenotypes. We also provide an ANOVA and a Kruskal-Wallis test between the phenotype and the six known covariates to uncover potential confounder effects (prior correction), which is displayed as a 'warning' table to inform the user about potential associations of the phenotype and any of the six known covariates. It is important to note that these ANOVA and Kruskal tests are conducted for informational purposes only, to assess potential associations between well-established inversions or symbiont infection status and the phenotype of interest. However, all known covariates are included in the model regardless, and PLINK2 will automatically correct for them, irrespective of the results from the ANOVA or Kruskal tests.

The interface also allows plotting an independent boxplot for each variant to visualize the effect of each allele on the phenotype. Importantly, for each variant, we also implemented a PheWAS button to visualize the effect of a particular variant across all existing phenotypes in DGRPool. We also annotated all the variants for impact (non-synonymous effects, stop-codon gain, etc.) and for potential regulatory effect (transcription factor binding motif disruption), which should assist researchers with prioritizing the variants in terms of potential consequences. For all of these variants, we provide links to their description in Flybase (*Gramates et al., 2022*).

As mentioned, these GWAS results are available for each existing phenotype in DGRPool, directly from the phenotype's page. But users can also submit their own phenotype files (through the 'Tool' menu in the header) and visualize the same information for their own phenotypes. The GWAS analysis runs in the backend and takes about 1–2 min before displaying the results. This is implemented using a queuing system which prevents overloading the server in case of a peak of users or requests.

After having run GWAS on all phenotypes in DGRPool, we observed the distribution of the number of significant variants per phenotype at $p \leq 1 \times 10^{-5}$ threshold, which is an often-used arbitrary threshold for GWAS analyses in DGRP studies (*Figure 4—figure supplement 2*). This threshold yields on average 118 significant hits per tested phenotype, which is skewed due to some phenotypes leveraging lots of results (median = 20). Interestingly, from the shape of the distribution, it seems that this threshold may not be sufficiently stringent for avoiding some false positives in the results. This may be specific to the GWAS parameters we used (`--quantile-normalize --variance-standardize`), which already provide a more stringent list of hits than without normalizing the phenotypes. However, in this configuration, the $p \leq 1 \times 10^{-6}$ threshold appears to be a good alternative for avoiding an excessive number of false positives (*Figure 4A*), as evident from the distribution of GWAS results across different thresholds (*Figure 4—figure supplement 2*). Another very often used threshold is the Bonferroni one, which is much more stringent and varies from $p \leq 1.126 \times 10^{-8}$ (if considering all 4 M variants) to $p \leq 2.67 \times 10^{-8}$ (if removing variants with low MAF or high number of missing values). In our results, the Bonferroni threshold ($p \leq 2.67 \times 10^{-8}$) yielded 20 significant hits on average (median = 0, *Figure 4—figure supplement 2*) which could be limiting for many studies as it may mask potentially interesting variants that, while minimally contributing on an individual basis, may collectively point to implicated pathways or biological processes (*Uffelmann et al., 2021*). Thus, while choosing an optimal threshold is in general challenging, our results indicate that any threshold below or equal to $1 \times 10^{-6}$ is reasonable given that at this threshold, the p-values appear not overestimated. We also verified if any variant is over-selected across all phenotypes to uncover a possible bias in our studies (*Figure 4B*). We found that only three loci are clearly overrepresented, corresponding to two studies with many correlated phenotypes (*Dembeck et al., 2015* and *Vonesch et al., 2016*; *Dembeck et al., 2015*; *Vonesch et al., 2016*). Both studies were already prominent in *Figure 2A*, both in terms of their number of phenotypes and

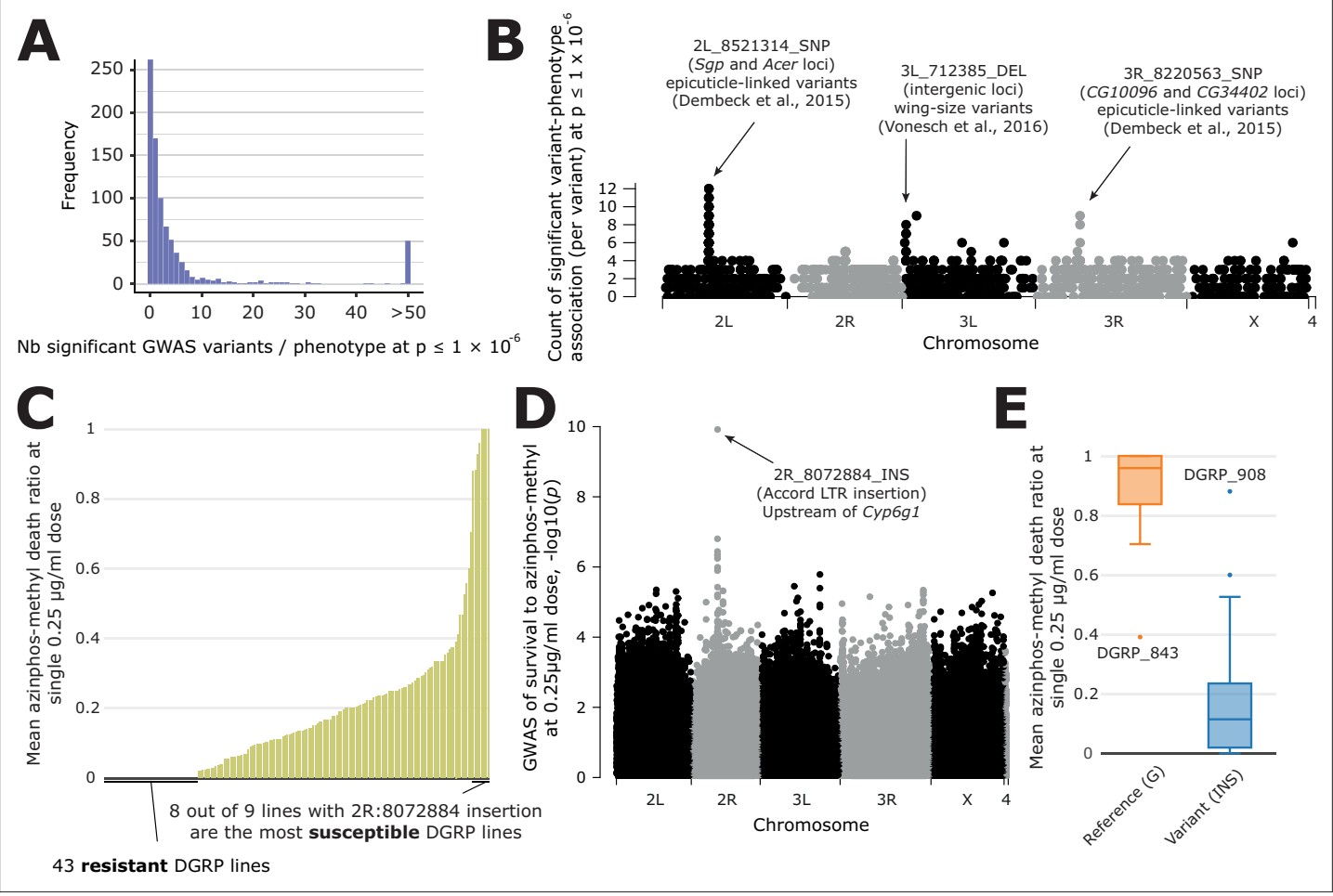

**Figure 4.** Overview of GWAS results across phenotypes and one case study. (**A**) Distribution of the number of significant variants after a GWAS, for each phenotype available in DGRPool. Of note, all values > 50 have been set to 50, for easier visualization. (**B**) For each variant, we plotted the number of times it was significantly associated with a phenotype (y-axis=number of occurrences). It is worth noting that we chose a Manhattan plot for representing this information, but this is not a 'real' GWAS Manhattan plot. (**C**) Case study on survival to azinphos-methyl exposure **Battlay et al., 2016**, here to a 0.25 µg/ml dose. This plot was extracted from the phenotype's page on DGRPool at https://dgrpool.epfl.ch/phenotypes/20. (**D**) Manhattan plot (taken from DGRPool's result page https://dgrpool.epfl.ch/phenotypes/20/gwas_analysis) showing the association of variants to the 'survival at 0.25 µg/ml dose' phenotype. (**E**) Boxplot (taken from DGRPool's result page https://dgrpool.epfl.ch/phenotypes/20/gwas_analysis), showing the effect of the top variant, 2 R:8072884, which is a long insertion.

The online version of this article includes the following figure supplement(s) for figure 4:

**Figure supplement 1.** Screenshot from the GWAS result page.

**Figure supplement 2.** Distribution of the number of GWAS hits per phenotype depending on the significance threshold.

high correlation. One study comprises 66 cuticular hydrocarbon composition traits (**Dembeck et al., 2015**) and the other involves 27 organismal size traits (**Vonesch et al., 2016**). These results are thus expected and do not indicate an overestimation of certain variants.

As a proof-of-concept and a validation of our approach, we compared our results with a randomly selected study that identified several variants associated with survival to azinphos-methyl at different doses (0.25, 0.5, 1, and 2 µg/ml; **Battlay et al., 2016**). Of note, this study is available in DGRPool under https://dgrpool.epfl.ch/studies/3. In particular, this study showed that survival to azinphos-methyl is highly variable among DGRP lines, even at a 'low'0.25 µg/ml dose. Importantly, the results of this study are reproduced in DGRPool as can be observed on the respective phenotype's page (https://dgrpool.epfl.ch/phenotypes/20, **Figure 4C**). For example, DGRPool's GWAS results are very similar to those of the study (https://dgrpool.epfl.ch/phenotypes/20/gwas_analysis, **Figure 4D**) and show a strong association at a 2R locus. Interestingly, the top variant we found, 2R:8072884 (p=1.966 x $10^{-26}$),

a 509 bp insertion polymorphism, is the *Accord* LTR insertion. It is annotated as located upstream of the *Cyp6g1* gene and has a high likelihood to be the main causal gene (*Daborn et al., 2002*; *Schmidt et al., 2010*). As described in the author's Ph.D. thesis (*Battlay, 2019*), the minor allele at this variant —which corresponds to NOT having the insertion— correctly genotypes eight out of nine susceptible DGRP lines that are homozygous for the ancestral *Cyp6g1*$^M$ arrangement at this locus (DGRP lines 091, 486, 642, 776, 802, 821, *843*, 852, and 857). The presence of the *Accord* LTR insertion is associated with increased resistance to organophosphates, suggesting that derived alleles of *Cyp6g1* confer organophosphate resistance in the DGRP (*Figure 4E*).

These results show that DGRPool is able to accurately reproduce results from existing studies, and that new biological findings can be leveraged from its interactive results and plots. Revisiting the same organophosphate study (*Battlay et al., 2016*), the PheWAS page present in the GWAS results shows that this top variant is not only significant at other doses, but that it is also significant in the context of another study investigating *Drosophila* microbiota (*Chaston et al., 2016*). This could help with fine-tuning putative causal variants, but also with uncovering potential associations between certain phenotypes that in turn could enable studies aimed at providing underlying genetic and molecular mechanisms.

## Extreme phenotypes

After having collected and harmonized thousands of DGRP phenotypes, we investigated if we could identify outliers amongst DGRP lines that would potentially bias phenotypic associations. Indeed, if a particular DGRP line is repeatedly ranked in the extreme of all phenotypes, it could be that there are unknown cofactors that make the line 'weaker' in general, or inversely. Although it is difficult to judge what phenotype is particularly advantageous or disadvantageous due to the presence of potential trade-offs (*Zwaan et al., 1995*; *Rose and Charlesworth, 1980*), we can determine how often a DGRP line is in the top *or* bottom 15% of a given phenotype. By focusing on phenotypes that are likely impacting overall viability, we ranked DGRP lines for each associated phenotype. Upon ranking the DGRP lines, we calculated whether the rank falls within the top or bottom 15% performers of the phenotype. We then assessed for each DGRP line how often they are 'extreme' and divided this by the total number of phenotypes in which the DGRP line has been included to obtain a 'fraction of extremeness' (FoE). Finally, we filtered for lines with at least 50 available phenotypic measures to ensure that our values were not driven by a low number of observations (*Figure 5A*). Overall, we observed a modest correlation in the FoE across the sexes (*Figure 5B*, Spearman's $\rho$ =0.3514, <1.57 x 10$^{-5}$). While this suggests that extremeness is a population-wide feature, the correlation is not sufficiently strong to conclude that DGRP lines are generally extreme in both sexes, which may only apply to specific lines.

Upon considering individual DGRP lines, we can observe to what extent they are extreme for each individual phenotype. In *Figure 5C*, we show the most extreme and 'moderate' (i.e. least distinctive) DGRP lines for each sex using an adjusted FoE for plotting purposes in which lower scores represent DGRP lines with a high FoE. While females of DGRP_879 and males of DGRP_783 tend to be extreme in some cases, for the majority of phenotypes, they are considered moderate. Conversely, females of DGRP_757 and males of DGRP_352 are more likely to be labeled as extreme.

These examples only represent extremeness for individual DGRP lines of a given sex, however, their counterpart may not be *as* extreme or moderate. We therefore also looked for DGRP lines which can be considered extreme in both females and males and are potentially more extreme on a population-wide basis. *Figure 5D* describes such populations where the overall FoE between males and females differed on average at most 0.05. In these cases, DGRP_852 and DGRP_042 are more likely to be extreme across sexes, which may be attributed to at least two factors. First, this may indicate that the population is generally not healthy if they consistently display a low lifespan, or second, and conversely, well-documented trade-offs of life history traits such as lifespan vs fecundity may be strongly at play here. The former does not however seem to be the case, as shown in *Figure 5E*. Both DGRP_852 and DGRP_042 generally display lifespan values around the mean lifespan of all DGRP lines, suggesting that they are more likely extreme for other phenotypes and are thus not by definition weak lines. However, DGRP_757 and DGRP_765 consistently display lower longevity in lifespan studies. These lines may therefore on the one hand be of particular interest for those studying life history traits in an evolutionary context, even though we did not observe strong lifespan and

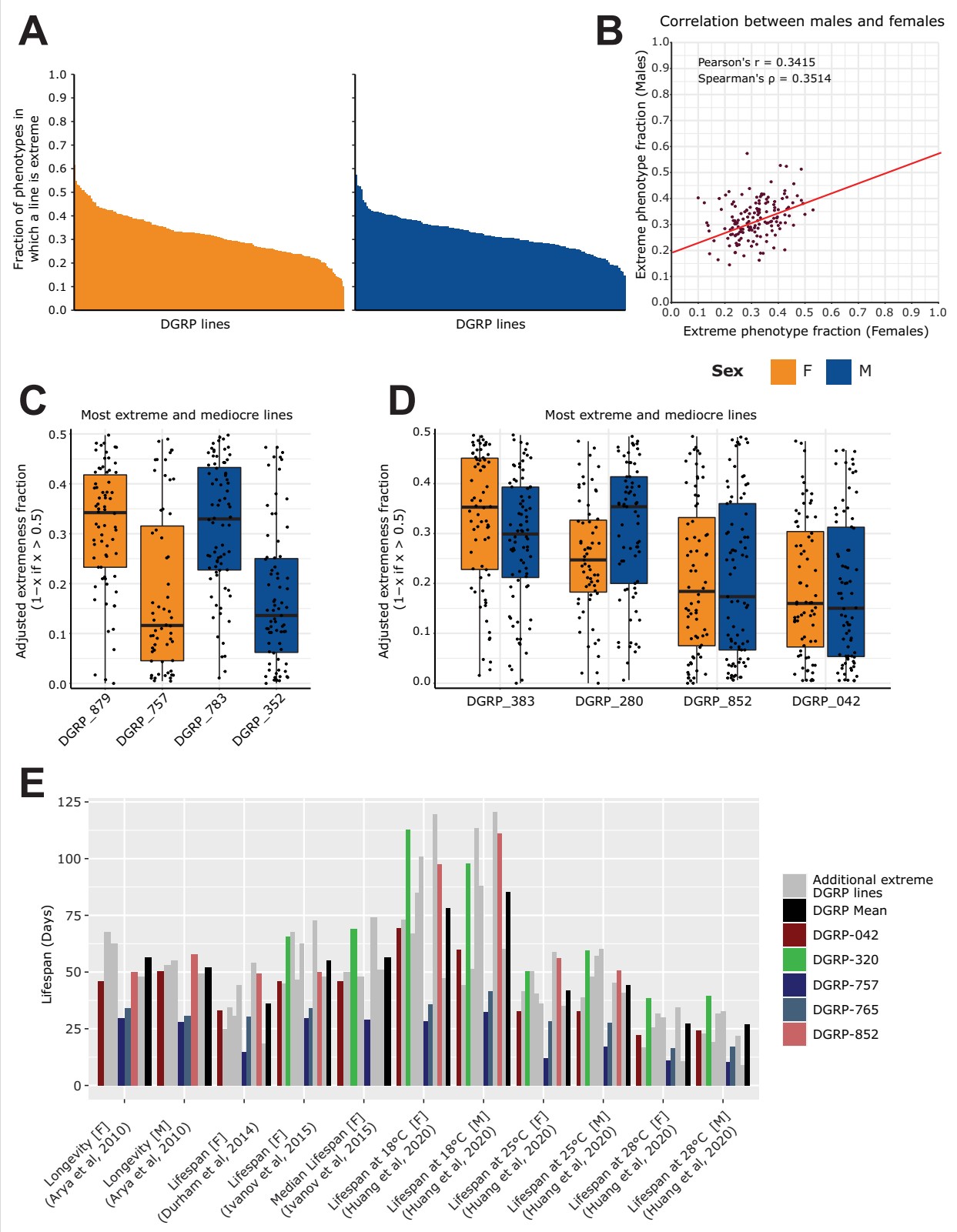

**Figure 5.** Analysis of extremeness among DGRP lines across 40 phenotypes. (**A**) Fraction of extremeness of a given DGRP line. DGRP lines are assigned as 'extreme' in a phenotype when they are in the top or bottom 15% of the phenotypic spectrum. Phenotypes were selected based on the curated studies which had the following categories assigned to them: Life history traits, Immunity, Toxicity, Resistance, Fecundity, Aging. DGRP lines were included if they had at least 50 phenotypic measures. (**B**) Scatter plot for the fraction of extremeness of DGRP lines. On the x-axis, the fraction of

*Figure 5 continued on next page*

*Figure 5 continued*

extremeness is plotted for females, whereas males are plotted on the y-axis. (**C**) Most extreme and moderate DGRP lines per sex. On the x-axis, the adjusted fraction of extremeness is provided. Individual fractions of extremeness per phenotype were retrieved for each DGRP line. The fraction was adjusted by 1 minus the fraction of extremeness if the fraction of extremeness was above 0.5. Because extremeness can range from 0 to 0.15 or 0.85–1, we adjusted the fraction of extremeness for plotting purposes. DGRP lines with a low adjusted fraction of extremeness are therefore more extreme, whereas a high adjusted fraction of extremeness is representative of more moderate DGRP lines. (**D**) Extreme and moderate DGRP line pairings. On the x-axis, the adjusted fraction of extremeness is provided. Extreme and moderate line pairings were retrieved by searching for DGRP lines for which the fraction of extremeness between females and males was not greater than 0.05 while still having the highest and lowest average fraction of extremeness (across sex). (**E**) Looking at phenotypes from *Figure 2D* marked as longevity/lifespan, for DGRP lines which are in the top 5 of fraction of (**E**) extremeness for each respective sex, including DGRP_852 and DGRP_042 (red shades) from **5D**. We specifically highlight DGRP_757, DGRP_765 in blue shades to show that they are across multiple studies in the lower end of the lifespan as is expected given that the lifespan trait is robust across studies. Similarly, DGRP_320 shows a trend in which it displays above average lifespan. Other extreme DGRP lines which were in each respective top 5 are displayed in gray.

fecundity trade-offs across our phenotype dataset. On the other hand though, it may be advisable not to include DGRP_757 and DGRP_765 when studying the genetic basis of these complex traits as their outlier status may not reflect common genetic principles.

In conclusion, this analysis showed that while certain lines exhibit lower longevity or outlier behavior for specific traits, we found no evidence of a general pattern of extremeness across all traits. Therefore, the data do not support the idea of 'super normals' or any other inherently biased lines that could significantly affect genetic studies.

## Discussion

There are many studies across organisms where collated phenotyping data has led to novel insights (*Greene et al., 2023*; *Doust et al., 2022*). Even though the *Drosophila* Genetic Reference Panel was formally released more than ten years ago, the resulting phenotype data of over 100 studies has so far not been combined into a single accessible resource. We anticipate that providing wider access to this data, as driven by FAIR principles (*Wilkinson et al., 2016*), will therefore facilitate our general understanding of the relationship between genotypes and phenotypes.

We have previously shown that using a subset of this resource effectively enabled us to establish a relationship between mitochondrial haplotypes and feeding behavior, which we experimentally validated (*Bevers et al., 2019*). Next to our own study, other studies have used a similar approach and compared their results to already published phenotypes. For example, *Wang et al., 2017* studied the resistance and tolerance of DGRP flies to the fungal pathogen *Metarhizium anisopliae* (Ma549) and found that the host's defense to Ma549 is correlated with its defense to the bacterium *Pseudomonas aeruginosa* (Pa14). But they also compared this result to several previously published DGRP phenotypes including oxidative stress sensitivity (*Jordan et al., 2012*), aggression (*Shorter et al., 2015*), nutritional scores (*Unckless et al., 2015*), sleep indices (*Harbison et al., 2013*), and others. Similarly, (*Zwarts et al., 2015*) studied the size of the cerebral cortex and the mushroom bodies (MB). They showed that these phenotypes are correlated with phenotypes from other studies like aggression (*Zwarts et al., 2011*) and sleep (*Harbison et al., 2013*). Therefore, we believe that DGRPool will either aid with validating the findings of a given study (i.e. higher bacterial resistance linked to overall resistance phenotypes) or by placing a study's phenotype data into a wider context (for example, linking brain size to behavioral phenotypes).

Moreover, having access to multiple studies studying similar phenotypes can also be of help for meta-analyses and increased statistical power. In the case of longevity for example, there are multiple studies that aggregated this phenotype, across similar or complementary DGRP lines. Therefore, one could conduct a meta-GWAS analysis (*Zeggini and Ioannidis, 2009*) by leveraging the replicates or combining the different lines into a single dataset. This tends to be a challenging process given the need for data harmonization and curation, which is exactly what we aimed to address by establishing DGRPool. Of course, since similar DGRP lines across laboratories still have the same genotype, they should not be treated as biological replicates, but phenotypes could be averaged across similar lines, which would reduce hidden covariates such as laboratory adaptation or batch effects. Moreover, complementary lines can be used to enhance power and potentially find

more small-effect associations. Indeed, researchers are increasingly advocating for collaboration and joining efforts to combine resources to enable more accurate, and reproducible results (*McCarthy et al., 2008*).

We recognize certain limitations of the current web tool, particularly the lack of eQTL or gene expression data integration. Properly integrating DGRP GWAS results with gene expression data in a fair and robust manner would require uniform processing of multiple public datasets, necessitating the cataloging and standardization of all available datasets through a consistent pipeline. Moreover, incorporating a 'cell type' or 'tissue' layer would be essential, as gene expression data from whole flies is not directly comparable to data from specific tissues or even specific conditions. Since phenotypes are often tissue-dependent, this information is vital. However, implementing these layers presented too big of a challenge and was beyond the scope of this paper.

Our data collection and harmonization efforts have already enabled us to conduct some interesting cross-study analyses, including an investigation into the presence of biases stemming from outlier DGRP lines. Our 'extremeness' analysis revealed that caution is warranted when selecting DGRP lines for specific studies, because, while some DGRP lines may be situated at the outer edge of the phenotypic spectrum by chance, DGRP_757 and DGRP_765 generally display lower lifespans in longevity studies. It is important to note that a shorter lifespan does not necessarily imply lower viability, as populations can still be propagated healthily. However, a shorter lifespan may also result from an impaired development (*May et al., 2015*) or developmental environment, which may confound the study of healthy aging (*Iliadi et al., 2012*). Consequently, researchers should consider excluding these extreme lines from their experimental designs to prevent loss of power or potential covariate biases. Furthermore, and beyond our current focus on DGRP lines, we may in the future also consider adding standard *D. melanogaster* lines such as w1118, YWB, YWN or ORB to DGRPool. This is because such lines have often been included as controls in DGRP studies (*Hoffman et al., 2021*), and for most of these, genomic information is also available.

Finally, in order to sustain the value of the DGRP as a resource and to promote more findings, we provide the following guidelines for future DGRP phenotyping studies:

- When available, report the raw datasets with values per fly. Optionally, but only in addition, the summary datasets can be provided, with values averaged across flies.
- Provide the data as a separate Excel or text file (TSV/CSV) in the form of a matrix, with DGRP lines in rows and phenotypes in columns. Avoid reporting the values in the form of a PDF or an image, because it complicates data extraction afterward.
- Clearly define the abbreviations in the tables and the units used for all phenotypes, so that the phenotyping dataset is self-explanatory and does not require an extended search in the main manuscript.
- Report all DGRP lines in the first column of the phenotyping file, and the corresponding sex in the second column (M, F, or NA), before all phenotypes. Be careful to use the same format for all DGRP lines (e.g. DGRP_XXX).
- Pick a common format for all *NA* values, whether reporting *NA*, or as an empty cell. But avoid mixing different formats.

In conclusion, we propose that DGRPool has two primary purposes within the *Drosophila* community and beyond. First, it can be used to evaluate potential associations between phenotypes and contribute to understanding the genetic architecture underlying complex traits. Second, it can serve as a catalyst for further research and inform broader validation experiments, as exemplified in our previous work (*Bevers et al., 2019*). In the latter study, the validation of our hypothesis would not have been feasible without a harmonized dataset of phenotype data, as the connection between mitochondrial haplotypes and food intake would have remained theoretical. To maximize the benefits of DGRPool, it should therefore remain subject to all FAIR principles, which unfortunately are still too often only implemented in terms of 'open' and 'sharing'. In other words, when large amounts of data are made publicly available without systematic curation or homogenization, data interoperability and reproducibility can be highly problematic. DGRPool is in this regard a crucial initial step towards making DGRP phenotyping data widely accessible and usable for the entire *Drosophila* research community.

## Methods

### Data availability

All phenotyping data aggregated in DGRPool can be downloaded in a common format on each phenotype page. In the 'Download' section on the front page, we also provide four.tsv files containing (1) All studies and their metadata (authors, citation,...), (2) All phenotypes and their metadata (name, description, unit, …), (3) All DGRP lines and their metadata (name, bloomington accession, …), and (4) a global file with all numerical phenotypes across all studies, formatted following our recommendations.

We also provide an API for programmatic access to the data hosted on our website. The API is described at this page: https://dgrpool.epfl.ch/home/api.

All codes used to produce the figures of this manuscript are available at this GitHub repository: https://github.com/DeplanckeLab/DGRPool, copy archived at *DeplanckeLab, 2024a*. The website code is available as a Docker container at this GitHub repository: https://github.com/DeplanckeLab/DGRPool_web, copy archived at *DeplanckeLab, 2024b*.

### Web application

The DGRPool web application is hosted on a virtual machine at EPFL. All compute-intensive calculations (i.e. GWAS) are performed on an HPC within EPFL and results are then moved to the virtual machine's local storage. The back-end is implemented with Ruby-on-Rails (RoR) 7 and all data is stored in a PostgreSQL relational database. The front-end uses different JavaScript libraries and is set to enable interactive usage. For instance, the application implements bootstrap tooltips to display HTML texts within tooltips, plotly.js v.2.16.1 to generate the scatter plots, bar plots and box plots, using *scattergl*, *bar* and *box* modes respectively, or Jquery autocomplete for phenotype search combined with a SOLR search engine running on the server side (used for the phenotype comparison tool).

### Semi-automated referencing of studies and/or phenotypes

To submit a new study, any user can submit a DOI from the front page. Then, all metadata associated with this study (authors, journal, date, …) are automatically imported from the Crossref API (*Hendricks et al., 2020*). When the study is created, it acquires the 'Submitted' state, and administrators are notified. Then, a curator is assigned to the study and needs to manually verify all information. A specific curator page allows him/her to (1) edit the metadata, (2) edit the categories associated with the study, or (3) add/remove/modify the phenotyping data and edit their names/types/units.

Identifiers from GEO (*Barrett et al., 2009*), ArrayExpress (*Brazma et al., 2003*), or the Sequence Read Archive (SRA) (*Kodama et al., 2011*) can be associated manually with any study, for example for referencing additional gene expression data that would be published along with the phenotyping data.

### GWAS

GWAS analyses (whether pre-calculated, or using the web tool) were computed using Plink2 v2.00a3LM (1 Jul 2021) with these parameters: "`--glm hide-covar --quantile-normalize --variance-standardize --geno 0.2 –maf 0.01`". It runs on all available variants in the DGRP database which is using the dm3 assembly (4,438,427 variants: 3,963,420 SNPs, 293,363 deletions, 169,053 insertions and 12,591 MNPs). We corrected the model for six known covariates (*Wolbachia* status, and five major insertions) that were described in *Huang et al., 2014* and also used on the DGRP2 website. Of note, these known covariates can also be considered as phenotypes, and thus are also available as a separate, browsable study on DGRPool (https://dgrpool.epfl.ch/studies/17). For each phenotype, we also provide (similar to the DGRP2 website), both a Kruskal-Wallis test and an ANOVA test to inform the user about the association between the phenotype and the six known covariates. We also provide the results of a Shapiro-Wilk normality test of the phenotype distribution. Of note, the Kruskal-Wallis test is used for a single factor (independent variables) at a time, unlike the ANOVA test which is handling multiple factors simultaneously (as it is performed in a multifactorial design).

### Extremeness

Fraction of extremeness was calculated for each phenotypic spectrum separately by ranking the values with ties being assigned the minimum rank. We then calculated a cut-off to assign ranks in the bottom

or upper 15% of a phenotypic range. This rank cut-off was further rounded up to be more inclusive on either end (i.e. if the cut-off was 1.2 or 1.8, the cut-off would become 2). Phenotypes equal or lower than the cut-off were assigned –1, whereas phenotypes equal to the max rank minus the cutoff or higher were assigned 1. Remaining phenotypic values were assigned 0. DGRP lines with phenotypic values of either –1 or 1 were then considered extreme for a given phenotype.

To calculate the overall fraction of extremeness for each DGRP line, we counted the number of times a DGRP line was assigned –1 or 1 and divided this by the total number of phenotypes available for that particular DGRP line. For most of our analyses, we only included DGRP lines for which at least 50 phenotypes were available unless stated otherwise.

The adjusted fraction of extremeness was calculated by dividing the phenotypic ranking by the max rank of a given phenotype. Values were adjusted with 1 minus the value if the value was above 0.5 (e.g. if x=0.91, the adjusted value is 1–0.91=0.09). Only adjusted fraction of extremeness values below 0.15 are therefore considered extreme. As no rounding was performed in this case, it is possible for DGRPs to be assigned –1 and labeled as extreme, even though the DGRP line may have a value of 0.167. Further analysis shows that this 'violation' only takes place for 1.1% (417 out 36,753) of the observations. At a *per* DGRP view, this would amount to less than 1 per 50 phenotypes, the cut-off for the number of phenotypes which a line needs to adhere to in order to be included in our analysis.

## Acknowledgements

The authors gratefully acknowledge the help and suggestions from Nathan M Fiorellino and Jasper Deplancke in the early stages of the development of this web tool. This work was funded by the Ecole Polytechnique Fédérale de Lausanne (EPFL) and SNSF Project Grant (#310030_197082) to BD.

## Additional information

### Funding

| Funder | Grant reference number | Author |
| --- | --- | --- |
| Swiss National Science Foundation Project Grant | #310030_197082 | Bart Deplancke |

The funders had no role in study design, data collection and interpretation, or the decision to submit the work for publication.

### Author contributions

Vincent Gardeux, Conceptualization, Resources, Data curation, Software, Formal analysis, Supervision, Validation, Visualization, Methodology, Writing – original draft, Project administration, Writing - review and editing; Roel PJ Bevers, Conceptualization, Formal analysis, Methodology, Writing – original draft; Fabrice PA David, Resources, Software, Formal analysis, Visualization, Methodology; Emily Rosschaert, Romain Rochepeau, Data curation, Formal analysis; Bart Deplancke, Conceptualization, Funding acquisition, Writing – original draft, Project administration, Writing - review and editing

### Author ORCIDs

Vincent Gardeux (ID) https://orcid.org/0000-0001-8954-2161
Bart Deplancke (ID) https://orcid.org/0000-0001-9935-843X

Reviewer #1 (Public Review): https://doi.org/10.7554/eLife.88981.3.sa1
Reviewer #2 (Public Review): https://doi.org/10.7554/eLife.88981.3.sa2
Author response https://doi.org/10.7554/eLife.88981.3.sa3

## Additional files

### Supplementary files
• Supplementary file 1. Spreadsheet containing all 135 studies. This table was created from the '

studies.tsv' file, which is downloadable from the front page of DGRPool. It contains all studies and publication references used in the online tool and in this manuscript.

• MDAR checklist

### Data availability

All phenotyping data aggregated in DGRPool can be downloaded in a common format on each phenotype page. In the 'Download' section on the front page, we also provide four .tsv files containing (1) All studies and their metadata (authors, citation, ...), (2) All phenotypes and their metadata (name, description, unit, ...), (3) All DGRP lines and their metadata (name, bloomington accession, ...), and (4) a global file with all numerical phenotypes across all studies, formatted following our recommendations. We also provide an API for programmatic access to the data hosted on our website. The API is described at this page: https://dgrpool.epfl.ch/home/api. All codes used to produce the figures of this manuscript are available at this GitHub repository: https://github.com/DeplanckeLab/DGRPool (copy archived at *DeplanckeLab, 2024a*). The website code is available as a Docker container at this GitHub repository: https://github.com/DeplanckeLab/DGRPool_web (copy archived at *DeplanckeLab, 2024b*).

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
