## [Editor Report · eLife Assessment]

Genetic analysis of complex traits in *Drosophila* provides a resource for exploring the relationship between genetic and phenotypic variation. The web tool DGRPool presented in this paper makes data and results from the *Drosophila* Genetic Reference Panel accessible that will enable downstream analyses of genetic association. The findings of this paper are considered to be **important**, with practical implications beyond a single subfield, supported by **convincing** evidence using appropriate and validated methodology in line with current state of the art.

---

## [Referee Report · Reviewer #1 (Public Review)]

This is a technically sound paper focused on a useful resource around the DRGP phenotypes which the authors have curated, pooled, and provided a user-friendly website. This is aimed to be a crowd-sourced resource for this in the future. The authors should make sure they coordinate as well as possible with the NC datasets and community and broader fly community.

---

## [Referee Report · Reviewer #2 (Public Review)]

In the present study, Gardeux et al provide a web-based tool for curated association mapping results from DRP studies. The tool lets users view association results for phenotypes and compare mean phenotype ~ phenotype correlations between studies. In the manuscript, the authors provide several example utilities associated with this new resource, including pan-study summary statistics for sex, traits, and loci. They highlight cross-trait correlations by comparing studies focused on longevity with phenotypes such as oxphos and activity. Strengths: -Considerable efforts were dedicated toward curating the many DRG studies provided. -Available tools to query large DRP studies are sparse and so new tools present appeal Weaknesses: The creation of a tool to query these studies for a more detailed understanding of physiologic outcomes seems underdeveloped. These could be improved by enabling usages such as more comprehensive queries of meta-analyses, molecular information to investigate given genes or pathways, and links to other information such as in mouse rat or human associations.

---

## [Author Response]

The following is the authors’ response to the original reviews.

We would like to thank the reviewers for their positive and constructive comments on the manuscript.

We committed in our original rebuttal letter to implement the following revisions to both DGRPool and the corresponding manuscript to address the reviewers’ comments:

(1) We agree with reviewer #1 that normalizing the data could potentially improve the GWAS results. Thus, for computing the GWAS results, we are now using these two additional options in PLINK2: “--quantile-normalize --variance-standardize”. We assessed the impact of these options on the overall results, which revealed only minor improvements of the results, globally being a bit more stringent. In this direction, we also now filter the top results with a nominal p-value of 0.001 instead of 0.01, also because it provided better results for the new gene set enrichment step.

(2) We added a KRUSKAL test next to the ANOVA test for assessing the links between the phenotypes and the 6 known covariates, as well as a Shapiro-Wilk test of normality.

(3) We agree with both reviewers that gene expression information is of interest. As mentioned before, adding gene expression data to the portal would have required extensive work, beyond the current scope of this paper, which primarily focuses on phenotypes and genotype-phenotype associations. Nonetheless, we included more gene-level outlinks to Flybase. Additionally, we now link variants and genes to Flybase's online genome browser, JBrowse. By following the reviewers' suggestions, we aim to guide DGRPool users to potentially informative genes.

(4) Consistent with the latter point, and in agreement with reviewer #2, we acknowledge that additional tools could enhance DGRPool's functionality and facilitate meta- analyses for users. Therefore, we developed a gene-centric tool that now allows users to query the database based on gene names. Moreover, we integrated ortholog databases into the GWAS results. This feature will enable users to extend *Drosophila* gene associations to other species if necessary.

(5) We amended the manuscript to describe all the new tools and features that were developed and implemented. In short, the new features include a new gene-centric page with diverse links (Phenotypes, Genome Browser JBrowse, Orthologs …), a variant-centric page (variant details, and PheWAS), an API for programmatic access to the database, and other statistical outputs and filtering options.

We will detail these advances in the point-by-point response below and in the revised manuscript.

**Reviewer #1 (Public Review):**
This is a technically sound paper focused on a useful resource around the DRGP phenotypes which the authors have curated, pooled, and provided a user-friendly website. This is aimed to be a crowd-sourced resource for this in the future.The authors should make sure they coordinate as well as possible with the NC datasets and community and broader fly community. It looks reasonable to me but I am not from that community.

We thank the reviewer for the positive comments. We will leverage our connections to the fly and DGRP communities to make the resource as valuable as possible. DGRPool in fact already reflects the input of many potential users and was also inspired by key tools on the DGRP2 website. Furthermore, it also rationalizes why we are bridging our results with other resources, such as linking out to Flybase, which is the main resource for the *Drosophila* community at large.

I have only one major concern which in a more traditional review setting I would be flagging to the editor to insist the authors did on resubmission. I also have some scene setting and coordination suggestions and some minor textual / analysis considerations.The major concern is that the authors do not comment on the distribution of the phenotypes; it is assumed it is a continuous metric and well-behaved - broad gaussian. This is likely to be more true of means and medians per line than individual measurements, but not guaranteed, and there could easily be categorical data in the future. The application of ANOVA tests (of the "covariates") is for example fragile for this.The simplest recommendation is in the interface to ensure there is an inverse normalisation (rank and then project on a gaussian) function, and also to comment on this for the existing phenotypes in the analysis (presumably the authors are happy). An alternative is to offer a kruskal test (almost the same thing) on covariates, but note PLINK will also work most robustly on a normalised dataset.

We thank the reviewer for raising this interesting point. Indeed, we did not comment on the distribution of individual phenotypes due to the underlying variability from one phenotype to another, as suggested by the reviewer. Some distributions appear normal, while others are clearly not normally distributed. This information is 'visible' to users by clicking on any phenotype; DGRPool automatically displays its global distribution if the values are continuous/quantitative. Now, we also provide a Shapiro-Wilk test to assess the normality of the distribution.

We acknowledge the reviewer's concerns regarding the use of ANOVA tests. However, we want to point out that the ANOVA test is solely conducted to assess whether any of the well- established inversions or symbiont infection status (that, for simplification, we call “covariates” or “known covariates”) are associated with the phenotype of interest. This is merely informational, to help the user understand if their phenotype of interest is associated with a known covariate. But all of these known covariates are put in the model in any case, so PLINK2 will automatically correct for them, whatever is the output of the ANOVA test.

Still, we amended the manuscript to better explain this, and we added a Kruskal-Wallis test (in addition to the ANOVA test) in the results, so the users can have a better overview of potentially associated known covariates. We added this text on p. 10 of the revised manuscript:

“The tool further runs a gene set enrichment analysis of the results filtered at p<0.001 to enrich the associated genes to gene ontology terms, and Flybase phenotypes. We also provide an ANOVA and a Kruskal-Wallis test between the phenotype and the six known covariates to uncover potential confounder effects (prior correction), which is displayed as a “warning” table to inform the user about potential associations of the phenotype and any of the six known covariates. It is important to note that these ANOVA and Kruskal tests are conducted for informational purposes only, to assess potential associations between well-established inversions or symbiont infection status and the phenotype of interest. However, all known covariates are included in the model regardless, and PLINK2 will automatically correct for them, irrespective of the results from the ANOVA or Kruskal tests. “

We also acknowledge in the manuscript (Methods section) that the Kruskal-Wallis test is used for a single factor (independent variables) at a time. This is unlike the ANOVA test that we initially performed, which was handling multiple factors simultaneously (given that it was performed in a multifactorial design). For a more direct comparison with our ANOVA model, we ran separate Kruskal-Wallis tests for each factor, but then we acknowledged its potential limitations compared to our multifactorial ANOVA, since each of these tests treats the factor in question as the only source of variation, not considering other factors. But since the test is not intended for interactions or combined effects of these factors, we deem it to be sufficient.

Nevertheless, we concur with the reviewer that normalizing the data could potentially enhance GWAS results. Consequently, we have rerun the GWAS analyses using the PLINK2 --quantile- normalize and --variance-standardize options. We have updated all results on the website and also updated the plots in the manuscript, accordingly.

Minor points:On the introduction, I think the authors would find the extensive set of human GWAS/PheWAS resources useful; widespread examples include the GWAS Catalog, Open Targets PheWAS, MR-base, and the FinnGen portal. The GWAS Catalog also has summary statistics submission guidelines, and I think where possible meta-data harmonisation should be similar (not a big thing). Of course, DRGP has a very different structure (line and individuals) and of course, raw data can be freely shown, so this is not a one-to-one mapping.

Thank you for the suggestion. We cited these resources in the Introduction.

“This aligns with the harmonization effort undertaken by other human GWAS/PheWAS resources, such as the GWAS Catalog, Open Targets PheWAS, MR-base, and the FinnGen portal, which provide extensive examples of effective data use and accessibility. Although the structure of DGRPool differs from these human databases, we acknowledge the importance of similar meta-data harmonization guidelines. Inspired by the GWAS Catalog's summary statistics submission guidelines, we propose submission guidelines for DGRP phenotyping data in this paper. “

For some authors coming from a human genetics background, they will be interpreting correlations of phenotypes more in the genetic variant space (eg LD score regression), rather than a more straightforward correlation between DRGP lines of different individuals. I would encourage explaining this difference somewhere.

We understand that this is a potential issue and we made the distinction clearer in the manuscript to avoid any confusion. We added this text on p.7, at the beginning of the correlation results section:

“Of note, by “phenotype correlations”, we mean direct phenotype-phenotype correlations, i.e. a straightforward Spearman’s correlation of two phenotypes between common DRGP lines, and we repeated this process for each pair of phenotypes. “

This leads to an interesting point that the inbred nature of the DRGP allows for both traditional genetic approaches and leveraging the inbred replication; there is something about looking at phenotype correlations through both these lenses, but this is for another paper I suspect that this harmonised pool of data can help.

We agree with the reviewer and hope that more meta-analyses will be made possible by leveraging the harmonized data that are made available through DGRPool.

I was surprised the authors did not crunch the number of transcript/gene expression phenotypes and have them in. Is this because this was better done in other datasets? Or too big and annoying on normalisation? I'd explain the rationale to leave these out.

This is a very good point and is in fact something that we initially wanted to do. However, to render the analysis fair and robust, it would require processing all datasets in the same way. This implies cataloging all existing datasets and processing them through the same pipeline. In addition, it would require adding a “cell type” or “tissue” layer, because gene expression data from whole flies is obviously not directly comparable to gene expression data from specific tissues or even specific conditions. This would be key information as phenotypes are often tissue-dependent. Consequently, and as implied by the reviewer, we deemed this too big of a challenge beyond the scope of the current paper. Nevertheless, we plan to continue investigating this avenue in a potential follow-up paper.

We still added a gene-centric tool to be able to query the GWAS results by gene. We also added orthologs and Flybase gene-phenotype information, both in this new gene-centric tool and also in all GWAS results.

I think 25% FDR is dangerously close to "random chance of being wrong". I'd just redo this section at a higher FDR, even if it makes the results less 'exciting'. This is not the point of the paper anyway.

We agree with the reviewer that this threshold implies a higher risk of false positive results. However, this is not an uncommonly used threshold (Li et al., PLoS biology, 2008; Bevers et al., Nature Metabolism, 2019; Hwangbo et al, Elife, 2023), and one that seems robust enough in our analysis since similar phenotypes are significant in different studies at different FDR thresholds.

Nevertheless, we revisited these results with a stronger threshold of 5% FDR in the main Figure 3C. Most of the conclusions were maintained, except for the relation between longevity and “food intake”, as well as “sleep duration”. We modified the manuscript accordingly, notably removing these points from the abstract, and tuning down the results section. We kept the 25% FDR results as supplemental information.

I didn't buy the extreme line piece as being informative. Something has to be on the top and bottom of the ranks; the phenotypes are an opportunity for collection and probably have known (as you show) and cryptic correlations. I think you don't need this section at all for the paper and worry it gives an idea of "super normals" or "true wild types" which ... I just don't think is helpful.

We appreciate the reviewer’s feedback on the section regarding extreme DGRP lines and understand the concern about potential implications of “super normals” or “true wild types.” This section aimed to explore whether specific DGRP lines consistently rank in the extremes of phenotypic measures, particularly those tied to viability-related traits. Our hypothesis was that if particular lines consistently appear at the top or bottom, this might suggest some inherent bias or inbreeding-related weakness that could influence genetic association studies.

However, as per the analyses presented, we did not discover support for this phenomenon. Importantly, the observed mild correlation in extremeness across sexes, while not profound, further suggested that this phenomenon is not a consistent population-wide feature.

Nevertheless, we consider that this message is still important to convey. In response to the reviewer's feedback, we have provided a clearer conclusion of this paper section by adding the following paragraph:

“In conclusion, this analysis showed that while certain lines exhibit lower longevity or outlier behavior for specific traits, we found no evidence of a general pattern of extremeness across all traits. Therefore, the data do not support the idea of 'super normals' or any other inherently biased lines that could significantly affect genetic studies. “

I'd say "well-established inversion genotypes and symbiot levels" rather than generic covariates. Covariates could mean anything. You have specific "covariates" which might actually be the causal thing.

We thank the author for the suggestion. We agree and modified the manuscript accordingly.

I wouldn't use the adjective tedious about curation. It's a bit of a value judgement and probably places the role of curation in the wrong way. Time-consuming due to lack of standards and best practice?

We thank the author for the suggestion. We agree and modified the manuscript accordingly, replacing the occurrences by “thorough” and “rigorous” which correspond better to the initial intended meaning.

**Reviewer #2 (Public Review):**
Summary:In the present study, Gardeux et al provide a web-based tool for curated association mapping results from DRP studies. The tool lets users view association results for phenotypes and compare mean phenotype ~ phenotype correlations between studies. In the manuscript, the authors provide several example utilities associated with this new resource, including pan-study summary statistics for sex, traits, and loci. They highlight cross-trait correlations by comparing studies focused on longevity with phenotypes such as oxphos and activity.Strengths:-Considerable efforts were dedicated toward curating the many DRG studies provided.-Available tools to query large DRP studies are sparse and so new tools present appealWeaknesses:The creation of a tool to query these studies for a more detailed understanding of physiologic outcomes seems underdeveloped. These could be improved by enabling usages such as more comprehensive queries of meta-analyses, molecular information to investigate given genes or pathways, and links to other information such as in mouse rat or human associations.

We appreciate the reviewer's kind comments.

Regarding the tools, we concur with the reviewer that incorporating additional tools could enhance DGRPool and facilitate users in conducting meta-analyses. Therefore, we developed two new tools: a gene-centric tool that enables users to query the database based on gene names, and a variant-centric tool mostly for studying the impact of specific genomic loci on phenotypes. Additionally, in all GWAS results, we added links to ortholog databases, thereby allowing users to extend fly gene associations to other species, if required.

Furthermore, we added links to the Flybase database, for variants, phenotypes, and genes that are already present in Flybase. We also link out to a 'genome browser-like' view (Flybase’s JBrowse tool) of the GWAS results centered around the affected variants/genes.

Finally, we now also perform a gene-set enrichment analysis for each GWAS result, both in the Flybase gene-phenotype database and the Gene Ontology (GO) database.

**Reviewer #2 (Recommendations For The Authors):**
(1) The authors discuss how current available DRG databases are basically data-dump sites and there is a need for integrative queries. Clearly, they spent (and are spending) considerable efforts into curating associations from available studies so the current resource seems to contain several areas of missed opportunities. The most clear addition would be to integrate gene-level queries. For example which genes underlie associations to given traits, what other traits map to a specific gene, or multiple genes which map to traits. This absence of integration is somewhat surprising given the lab's previous analyses of eQTL data in DRPs (https://doi.org/10.1371/journal.pgen.1003055) and readily available additional data (ex. 10.1101/gr.257592.119 ,flybase) simple intersections between these at the locus level would provide much deeper molecular support for searching this database.

The point raised by the reviewer concerning eQTL / transcriptomic data is in fact similar to the one raised by reviewer #1. We strongly agree with both reviewers that incorporating eQTL results in the tool would be very valuable, and this is in fact something that we initially wanted to do. However, to render the analysis fair and robust, it would require re-processing multiple public datasets in the same way. This would imply cataloging all existing datasets and processing them through the same pipeline. In addition, it would require adding a “cell type” or “tissue” layer, because gene expression data from whole flies is obviously not directly comparable to gene expression data from specific tissues or even specific conditions. This would be key information as phenotypes are often tissue-dependent. Consequently, we deemed implementing all these layers too big of a challenge beyond the scope of the current paper, but we plan to continue investigating this avenue in a potential follow-up paper.

As mentioned before, we still integrated gene-level queries in a new tool, querying genes in the context of GWAS results. We acknowledge that this is not directly related to gene expression, and thus not implicating eQTL datasets (at least for now), but we think that it is for now a good alternative, reinforcing the interpretation of the GWAS results.

Since this point was raised by both reviewers, we added a discussion about this in the manuscript.

“We recognize certain limitations of the current web tool, particularly the lack of eQTL or gene expression data integration. Properly integrating DGRP GWAS results with gene expression data in a fair and robust manner would require uniform processing of multiple public datasets, necessitating the cataloging and standardization of all available datasets through a consistent pipeline. Moreover, incorporating a “cell type” or “tissue” layer would be essential, as gene expression data from whole flies is not directly comparable to data from specific tissues or even specific conditions. Since phenotypes are often tissue-dependent, this information is vital. However, implementing these layers presented too big of a challenge and was beyond the scope of this paper. “

(2) Another area that would help to improve is to provide either a subset or the ability to perform a meta-analysis of the studies proposed to see where phenotype intersections occur, as opposed to examining their correlation structure. For any given trait the PLINK data or association results seem already generated so running together and making them available seems fairly straightforward. This can be done in several ways to highlight the utility (for example w/wo specific covariates from Huang et al., 2014 and/or comparing associations that occur similarly or differently between sexes).

We are not 100% sure what the reviewer refers to when mentioning “phenotype intersection”, but we interpreted it as a “PheWAS capability”. Currently, in DGRPool, for every variant, there is a PheWAS option, which scans all phenotypes across all studies to see if several phenotypes are impacted by this same variant.

We tried to make this tool more visible, both in the GWAS section of the website, but also in the “Check your phenotype” tool, when users are uploading their own data to perform a GWAS. We have also created a “Variants” page, accessible from the top menu, where users can view particular variants and explore the list of phenotypes they are significantly associated with.

From both result pages, users can download the data table as .tsv files.

(3) As pointed out by the authors, an advantage of DRGs is the ease of testing on homozygous backgrounds. For each phenotype queried (or groups of related phenotypes would be of interest too), I imagine that subsetting strains by the response would help to prioritize lines used for follow-up studies. For example, resistant or sensitive lines to a given trait. This is already done in Fig 4C and 4E but should be an available analysis for all traits.

For all quantitative phenotypes, we show the global distribution by sex, followed by the sorted distribution by DGRP line. Since the data can be directly downloaded from the corresponding plots, resistant and sensitive lines can then be readily identified for all phenotypes.

(4) To researchers beyond the DRP community, one feature to consider would be seeing which other associations are conserved across species. While doing this at the phenotype level might be tricky to rename, assigning gene-level associations would make this streamlined. For example, a user could query longevity, subset by candidate gene associations then examine outputs for what is associated with orthologue genes in humans (ex. https://www.ebi.ac.uk/gwas/docs/file-downloads) or other reference panels such as mice and rats.

In all GWAS results, and in the gene-centric tool, we have added links to ortholog databases. In short, when clicking on a variant, users can see which gene is potentially impacted by this variant (gene-level variant annotation). When clicking on these genes, the user can then open the corresponding, detailed gene page.

To address the reviewer’s comment, in the gene page, we have added two orthologous databases (Flybase and OrthoDB), which enables cross-species association analyses.

(5) Related to enabling a meta-data analysis, it would be helpful to let users download all PLINK or DGRP tables in one query. This would help others to query all data simultaneously.

We would like to kindly point out that all phenotyping data can already be downloaded from the front page, which includes the phenotypes, the DGRP lines and the studies’ data and metadata. However, we did not provide the global GWAS results through a single file, because the data is too large. Instead, we provide each GWAS dataset via a unique file, available per phenotype, on the corresponding GWAS result page of this phenotype. This file is filtered for p<0.001, and contains GWAS results (PLINK beta, p and FDR) as well as gene and regulatory annotations.

(6) Following analysis of association data an interesting feature would be to enable users to subset strains for putative LOF variants at a given significant locus. This is commonly done for mouse strains (ex. via MGI).

The GWAS result table available for each phenotype can be filtered for any variant of interest. We added the capability to filter by variant impact; LOF variants being usually referred to as HIGH impact variants.

(7) Viewing the locus underlying annotation can also provide helpful information. For example, several nice fly track views are shown in 10.1534/g3.115.018929, which would help users to interpret molecular mechanisms.

We now link the GWAS results out to Flybase’s JBrowse genome browser.